

# A comparison of chloroplast genome sequences in *Aconitum* (Ranunculaceae): a traditional herbal medicinal genus

Hanghui Kong[1,4,*], Wanzhen Liu[2,*], Gang Yao[3] and Wei Gong[2]

[1] Key Laboratory of Plant Resources Conservation and Sustainable Utilization, South China Botanical Garden, Chinese Academy of Sciences, Guangzhou, China
[2] College of Life Sciences, South China Agricultural University, Guangzhou, China
[3] College of Forestry and Landscape Architecture, South China Agricultural University, Guangzhou, China
[4] Guangdong Provincial Key Laboratory of Applied Botany, South China Botanical Garden, Chinese Academy of Sciences, Guangzhou, China
[*] These authors contributed equally to this work.

Corresponding author
Wei Gong, wgong@scau.edu.cn

## ABSTRACT

The herbal medicinal genus *Aconitum* L., belonging to the Ranunculaceae family, represents the earliest diverging lineage within the eudicots. It currently comprises of two subgenera, *A.* subgenus *Lycoctonum* and *A.* subg. *Aconitum*. The complete chloroplast (cp) genome sequences were characterized in three species: *A. angustius*, *A. finetianum*, and *A. sinomontanum* in subg. *Lycoctonum* and compared to other *Aconitum* species to clarify their phylogenetic relationship and provide molecular information for utilization of *Aconitum* species particularly in Eastern Asia. The length of the chloroplast genome sequences were 156,109 bp in *A. angustius*, 155,625 bp in *A. finetianum* and 157,215 bp in *A. sinomontanum*, with each species possessing 126 genes with 84 protein coding genes (PCGs). While genomic rearrangements were absent, structural variation was detected in the LSC/IR/SSC boundaries. Five pseudogenes were identified, among which $\Psi rps$19 and $\Psi ycf$ 1 were in the LSC/IR/SSC boundaries, $\Psi rps$16 and $\Psi inf$A in the LSC region, and $\Psi ycf$ 15 in the IRb region. The nucleotide variability (*Pi*) of *Aconitum* was estimated to be 0.00549, with comparably higher variations in the LSC and SSC than the IR regions. Eight intergenic regions were revealed to be highly variable and a total of 58–62 simple sequence repeats (SSRs) were detected in all three species. More than 80% of SSRs were present in the LSC region. Altogether, 64.41% and 46.81% of SSRs are mononucleotides in subg. *Lycoctonum* and subg. *Aconitum*, respectively, while a higher percentage of di-, tri-, tetra-, and penta-SSRs were present in subg. *Aconitum*. Most species of subg. *Aconitum* in Eastern Asia were first used for phylogenetic analyses. The availability of the complete cp genome sequences of these species in subg. *Lycoctonum* will benefit future phylogenetic analyses and aid in germplasm utilization in *Aconitum* species.

## INTRODUCTION

The chloroplast (cp) is an intracellular organelle that plays an important role in the process of photosynthesis and it is widely present in algae and plants (*Neuhaus & Emes, 2000*; *Inoue, 2011*). The cp genome in angiosperms is a circular DNA molecule with a typically quadripartite structure, consisting of two copies of a large inverted repeat (IR) region that separates a large-single-copy (LSC) region from a small-single-copy (SSC) region (*Raubeson & Jansen, 2005*; *Yang et al., 2010*; *Green, 2011*; *Wicke et al., 2011*). Although highly conserved among plants, some differences in gene synteny, copy number and pseudogenes have been observed in cp genome structures (*Roy et al., 2010*; *Lei et al., 2016*; *Ivanova et al., 2017*). A complete cp genome is valuable for plant taxonomical analyses, phylogenetic reconstructions, speciation processes, and biogeographical inferences at different taxonomic levels. The cp genome is useful in investigating the maternal origin in plants, especially those with polyploid species, due to their haploid maternal inheritance and high conservation in gene content and genome structure (*Birky, 1995*; *Soltis & Soltis, 2000*; *Song et al., 2002*). High-throughput sequencing technologies have enabled a rapid increase in the completion of cp genomes and have shifted the study of phylogenetics to phylogenomics. Highly informative universal markers based on indels, substitutions, and inversions of the cp genome have been further developed for various molecular studies in plants.

The genus *Aconitum* L. belongs to the tribe Delphinieae in the Ranunculaceae family and represents one of the earliest diverging lineages within the eudicots APG IV (*Wang et al., 2009*; *Sun et al., 2011*; *The Angiosperm Phylogeny Group et al., 2016*). It is currently divided into two subgenera, *A.* subgenus *Lycoctonum* and *A.* subgenus *Aconitum*, comprising about more than 400 species throughout Eurasia and North America with its diversification center in Eastern Asia (*Utelli, Roy & Baltisberger, 2000*; *Jabbour & Renner, 2012*; *Wang et al., 2013*). Polyploid species were identified in both subgenera, particularly in subg. *Lycoctonum*. One of the tetraploid species in subg. *Lycoctonum* is *A. angustius* ($2n = 4x = 32$), which possesses heterologous chromosomes and is hypothesized to be a hybrid of *A. finetianum* ($2n = 2x = 16$) and *A. sinomontanum* ($2n = 2x = 16$) (*Gao, 2009*; *Kong et al., 2017b*). The three species display intermediate morphological characteristics and overlapping geographical distributions (*Shang & Lee, 1984*; *Yuan & Yang, 2006*; *Gao, 2009*; *Gao, Ren & Yang, 2012*). Based on previous morphological analysis and phylogenetic inference, *A. finetianum* was inferred to be the putative maternal progenitor of *A. angustius* (*Gao, 2009*; *Kong et al., 2017b*).

The genus *Aconitum* is known as a taxonomically and phylogenetically challenging taxon. Early divergence between subg. *Lycoctonum* and subg. *Aconitum* in Europe was suggested based on *trn*H-*psb*A and ITS (*Utelli, Roy & Baltisberger, 2000*). Although high morphological variability within and among populations was detected due to recent speciation, the morphological characteristics are poor indicators of relatedness. *Jabbour & Renner (2012)* conducted a phylogenetic reconstruction focusing on Delphineae based on *trn*L-F and ITS that suggested *Aconitum* was monophyletic clade and a sister group of *Delphinium*. However, few species from Eastern Asia were used, which may have affected

the previous phylogenetic analysis. Most recently, phylogenetic inferences of polyploid species relationships in subg. *Lycoctonum* were made using four cpDNA intergenic regions (*ndh*F-*trn*L, *psb*A-*trn*H, *psb*D-*trn*T, and *trn*T-L) and two nrDNA regions (ITS and ETS) (*Kong et al., 2017b*), *Aconitum finetianum* was inferred as the maternal progenitor of *A. angustius*. With the same cpDNA intergenic regions, taxonomical revision has been conducted based on phylogenetic analyses of subg. *Lycoctonum* by *Hong et al. (2017)*, yet phylogenetic information at the genomics level has been absent.

Although some *Aconitum* species are highly toxic because of aconite alkaloid, many species are essential in the formulation of traditional herbal medicine in Asia (*Zhao et al., 2010*; *Semenov et al., 2016*; *Wada et al., 2016*; *Liang et al., 2017*). The current state of *Aconitum* phylogenetics lacks molecular information of some species in Eastern Asia, and thus inhibits identification and germplasm utilization of this genus. In this study, we report the complete cp genome sequences of three species in subg. *Lycoctonum*; we established and characterized the organization of the cp genome sequences of tetraploid *A. angustius* as well as diploid *A. finetianum* and *A. sinomontanum*. We further compared the structure, gene arrangement and microsatellite repeats (SSRs) with the related species in both subgenera of *Aconitum*. Altogether, 14 species and two varieties from *Aconitum* were used for phylogenetic reconstruction at the genomic level. Seven previously unanalyzed species from the subg. *Aconitum* in Eastern Asia were investigated for phylogenetic relationships, and the maternal origin of *A. finetianum* was explored in the tetraploid, *A. angustius*. Our results provide cp genomic information for taxonomical identification, phylogenetic inference, or the population history of *Aconitum* or Ranunculaceae, which can also aid in the utilization of the genetic resources of *Aconitum* as a traditional herbal medicine.

## MATERIALS AND METHODS

### Plant samples and DNA extraction
Fresh leaves were collected from *A. angustius*, *A. finetianum* and *A. sinomontanum* growing in the greenhouse of South China Botanical Garden, Chinese Academy of Sciences. Total genomic DNA was extracted from the fresh leaves of *A. angustius*, *A. finetianum* and *A. sinomontanum* using the modified CTAB method (*Doyle & Doyle, 1987*). The DNA concentration was quantified using a Nanodrop spectrophotometer (Thermo Scientific, Carlsbad, CA, USA), and a final DNA concentration of >30 ng/μL was used for Illumina sequencing.

### Chloroplast genome sequencing, assembly and annotation
We sequenced the complete cp genome of *A. angustius*, *A. finetianum* and *A. sinomontanum* with an Illumina HiSeq 2000 at Beijing Genomics Institute (BGI) in Wuhan, China. Genomic DNA was fragmented randomly and then the required length of DNA fragments was obtained by electrophoresis. Adapters were ligated to DNA fragments followed by cluster preparation and sequencing. A paired-end library was constructed with 270 bp insert size, and then 150 bp paired reads were sequenced using an Illumina HiSeq 2000.

We assembled the cp genomes using Geneious 9.1.4 (Biomatters Ltd., Auckland, New Zealand) with BLAST and map reference tools, respectively. Using the program

DOGMA (http://dogma.ccbb.utexas.edu/) (*Wyman, Jansen & Boore, 2004*) and Geneious, annotation was performed in comparison with the cp genomes of *A. reclinatum* (MF186593) (*Kong et al., 2017a*), *A. barbatum* var. *puberulum* (KC844054) (*Chen et al., 2015*), and *A. barbatum* var. *hispidum* (KT820664) in subg. *Lycoctonum* as well as 10 species from the subg. *Aconitum* (*Choi et al., 2016*; GB Kim, CE Lim & JH Mun, 2016, unpublished data; *Lim et al., 2017*; D Yang, 2016, unpublished data; SG Yang et al., 2017, unpublished data) (Table 1). Altogether, 14 species and two varieties in both subgenera of *Aconitum* were used for annotation (Table 1). Among those species, *A. angustius*, *A. finetianum*, *A. sinomontanum*, *A. barbatum* var. *hispidum*, and *A. barbatum* var. *puberulum* were collected from China (*Chen et al., 2015*), *A. reclinatum* came from the United States (*Kong et al., 2017a*), while the remaining species were all sampled from Korea (*Choi et al., 2016*; GB Kim, CE Lim & JH Mun, 2016, unpublished data; *Lim et al., 2017*; D Yang, 2016, unpublished data; SG Yang et al., 2017, unpublished data). Chloroplast genome sequences of *Aconitum* species from Europe were not available in GenBank.

The annotation of tRNA genes were confirmed using the ARAGORN program (*Laslett & Canback, 2004*), and then manually adjusted using the program Geneious. Contigs with BLAST hits to consensus sequence from the "map to reference function" were assembled manually to construct complete chloroplast genomes. Finally, the circular genome maps of the three species were illustrated using the Organellar Genome DRAW tool (OGDRAW, http://ogdraw.mpimp-golm.mpg.de/) (*Lohse et al., 2013*). The annotated chloroplast genomic sequences of *A. angustius*, *A. finetianum* and *A. sinomontanum* have been submitted to GenBank (Accession Number: MF155664, MF155665 and MF155666).

## Genome comparison and divergence hotspot

The cp genome sequences from the finalized data set were aligned with MAFFT v7.0.0 (*Katoh & Frith, 2012*) and adjusted manually when necessary. Based on many other cp genome studies, the IRs expansion/contraction could lead to changes in the structure of the cp genome, leading to the length variation of angiosperm cp genomes and contributing to the formation of pseudogenes (*Kim & Lee, 2004*; *Nazareno, Carlsen & Lohmann, 2015*; *Ivanova et al., 2017*). Therefore, we conducted comparative analysis to detect the variation in the LSC/IR/SSC boundaries among the species/varieties. Comparative analysis of the nucleotide diversity (*Pi*) among the complete cp genomes of *Aconitum* was performed based on a sliding window analysis using DnaSP 5.10 (*Librado & Rozas, 2009*). The window length was 600 bp and step size was 200 bp. To test and visualize the presence of genome rearrangement and inversions, gene synteny was performed using MAUVE as implemented in Geneious with default settings based on 14 species and two varieties in both subgenera.

## Simple sequence repeats analysis

MISA (http://pgrc.ipk-gatersleben.de/misa/misa.html) (*Thiel et al., 2003*) is a tool for the identification and location of perfect microsatellites and compound microsatellites (two individual microsatellites, disrupted by a certain number of bases). We used MISA to search for potential simple sequence repeats (SSRs) loci in the cp genomes of the three species. The minimum number (thresholds) of SSRs was set as 10, 5, 4, 3, and 3 for
**Table 1** Summary of characteristics in chloroplast genome sequences of thirteen species and two varieties in *Aconitum*.

| | GenBank no. | Voucher number/herbarium | Total genome size (bp) | LSC (bp) | SSC (bp) | IR (bp) | Total number of genes | Protein-coding genes | tRNA genes | rRNA genes | GC content |
|---|---|---|---|---|---|---|---|---|---|---|---|
| **subg. *Lycoctonum*** | | | | | | | | | | | |
| *A. angustius* | MF155664 | ZY37/IBSC | 156,109 | 86,719 | 16,914 | 26,225 | 126 | 84 | 38 | 4 | 38% |
| *A. finetianum* | MF155665 | ZY25/IBSC | 155,625 | 86,664 | 17,107 | 25,927 | 126 | 84 | 38 | 4 | 38% |
| *A. sinomontanum* | MF155666 | ZY46/IBSC | 157,215 | 88,074 | 16,926 | 26,090 | 126 | 84 | 38 | 4 | 38% |
| *A. reclinatum* | MF186593 | US17/IBSC | 157,354 | 88,269 | 16,963 | 26,061 | 127 | 86 | 37 | 4 | 38% |
| *A.barbatum* var. *puberulum* | KC844054 | Not provided/- | 156,749 | 87,630 | 16,941 | 26,089 | 127 | 85 | 38 | 4 | 38% |
| *A.barbatum* var. *hispidum* | KT820664 | VP0000486327/NIBR | 156,782 | 87,661 | 16,987 | 26,067 | 127 | 85 | 38 | 4 | 38% |
| **subg. *Aconitum*** | | | | | | | | | | | |
| *A. austrokoreense* | KT820663 | VP0000494173/NIBR | 155,682 | 86,388 | 17,054 | 26,120 | 126 | 83 | 39 | 4 | 38.1% |
| *A.carmichaelii* | KX347251 | ACAR20151205/- | 155,737 | 86,330 | 17,021 | 26,193 | 124 | 83 | 37 | 4 | 38.1% |
| *A. chiisanense* | KT820665 | VP0000494177/NIBR | 155,934 | 86,559 | 17,085 | 26,145 | 125 | 82 | 39 | 4 | 38.1% |
| *A. ciliare* | KT820666 | VP0000486323/NIBR | 155,832 | 86,452 | 17,084 | 26,148 | 126 | 83 | 39 | 4 | 38.1% |
| *A. coreanum* | KT820667 | VP0000486326/NIBR | 157,029 | 87,622 | 17,035 | 26,186 | 128 | 86 | 38 | 4 | 38.0% |
| *A. jaluense* | KT820669 | VP0000494219/NIBR | 155,926 | 86,406 | 17,090 | 26,215 | 126 | 83 | 39 | 4 | 38.1% |
| *A. japonicum* | KT820670 | VP0000494223/NIBR | 155,878 | 86,480 | 17,104 | 26,147 | 127 | 84 | 39 | 4 | 38.1% |
| *A. kusnezoffii* | KT820671 | VP0000529885/NIBR | 155,862 | 86,335 | 17,103 | 26,212 | 126 | 84 | 39 | 4 | 38.1% |
| *A. monanthum* | KT820672 | VP0000529886/NIBR | 155,688 | 86,292 | 16,996 | 26,200 | 125 | 82 | 39 | 4 | 38.1% |
| *A. volubile* | KU556690 | MBC_KIOM-2015-73/KIOM | 155,872 | 86,348 | 16,944 | 26,290 | 126 | 83 | 38 | 4 | 38.1% |

mono-, di-, tri-, tetra-, and penta-nucleotides SSRs, respectively. All of the repeats found were manually verified and the redundant ones were removed.

## Phylogenetic analysis

Four species and two varieties in subg. *Lycoctonum* and 10 species in subg. *Aconitum* were used for phylogenetic reconstruction, with *Megaleranthis saniculifolia* and *Clematis terniflora*v as the outgroup. Except for *A. kusnezoffii*, *A. volubile*, and *A. ciliare*, the remaining seven species in subg. *Aconitum* from Korea were first used for phylogenetic analysis. The complete cp genome sequences and PCGs were used for the phylogenetic reconstruction of *Aconitum* species in Eastern Asia. Three different methods including Bayesian Inference (BI), Maximum Parsimony (MP), and Maximum Likelihood (ML) were employed. In all analyses, gaps were treated as missing.

Bayesian Inference (BI) of the phylogenies was performed using MrBayes v.3.2 (*Huelsenbeck & Ronquist, 2001*; *Ronquist & Huelsenbeck, 2003*). The best model was determined for each sequence partition, after comparisons among 24 models of nucleotide substitution using Modeltest v.3.7 (*Posada & Crandall, 1998*). We performed MP using PAUP* v.4.0b10 (*Swofford, 2002*). We calculated the bootstrap values with 1,000 bootstrap replicates, each with 10 random sequence addition replicates holding a single tree for each run. We conducted ML using RAxML (*Stamatakis, 2006*) and the RAxML graphical interface (raxmlGUI v.1.3 (*Silvestro & Michalak, 2012*)) with 1,000 rapid bootstrap replicates. The general time-reversible (GTR) model was chosen with a gamma model for the rate of heterogeneity.

# RESULTS AND DISCUSSION

## Genome organization and features

Using the Illumina HiSeq 2000 sequencing platform, a total number of $2 \times 150$ bp pair-end reads ranging from 9,879,068 to 27,530,148 bp were produced for the three species in subg. *Lycoctonum*. Altogether, 1,270 Mb of clean data were produced for *A. angustius*, 3,586 Mb for *A. finetianum,* and 3,590 Mb for *A. sinomontanum*. The assembly generated an average of 6,713 contigs with a N50 length of 732 bp for *A. angustius*, an average of 6,201 contigs with a N50 length of 801 bp bp for *A. finetianum*, and an average of 6,999 contigs with a N50 length of 769 bp for *A. sinomontanum*. Scaffolds from the assembly with k-mer values of 35 to 149 were matched to reference cp genome sequences, which were used to determine the relative position and direction respectively. We generated a new draft chloroplast genome by manually identifying the overlapping regions. To further refine the draft genome, the quality and coverage of each was double-checked by remapping reads. The complete cp genome sequences of the three species with full annotations were deposited into GenBank.

The size of the cp genomes was 156,109 bp for *A. angustius*, 155,625 bp for *A. finetianum* and 157,215 bp for *A. sinomontanum* (Table 1). The chloroplast genomes displayed a typical quadripartite structure, including a pair of IRs (25,927–26,225 bp) separated by LSC (86,664–88,074 bp) and SSC (16,914–17,107 bp) regions (Fig. 1 and Table 1). The GC content of the three cp genomes was 38.00%, demonstrating congruence with other *Aconitum* species (38.00% or 38.10%) (Table 1).

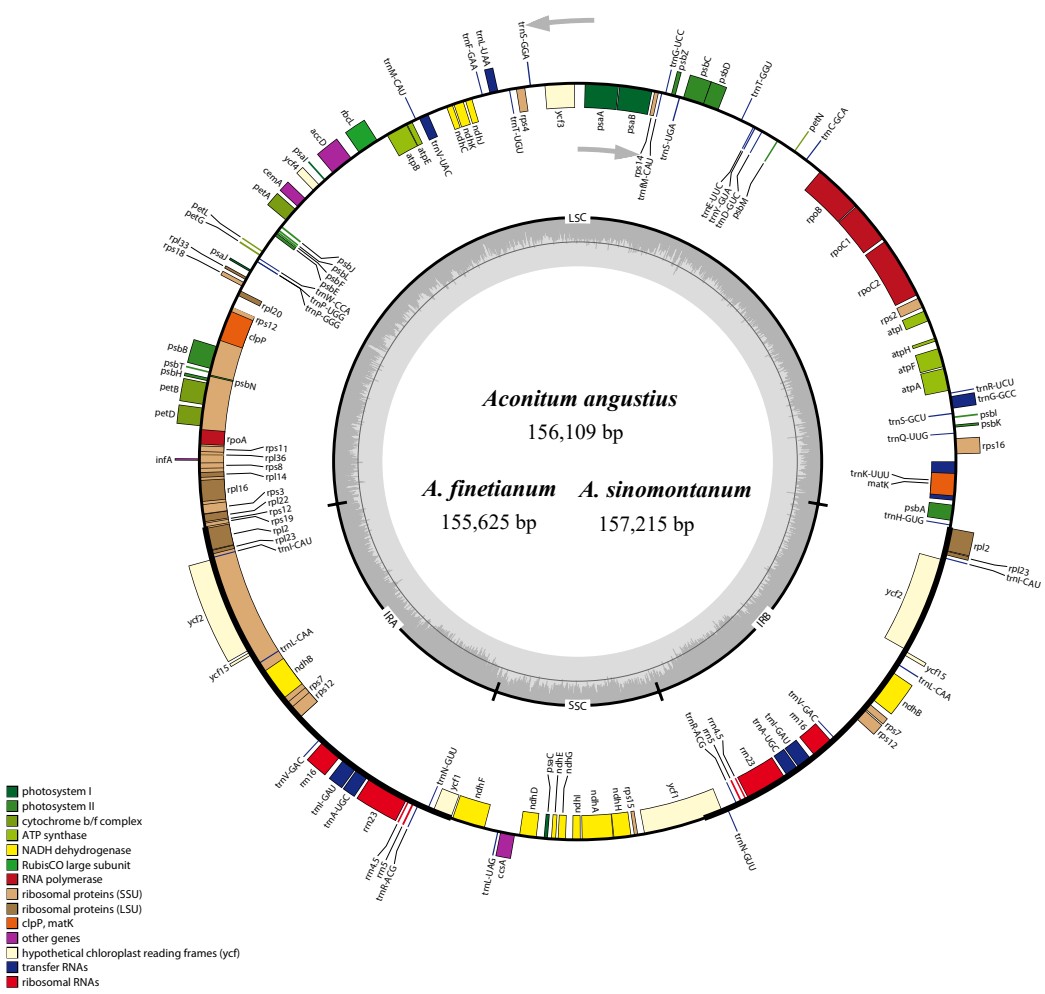

**Figure 1** **The gene maps of *Aconitum angustius*, *A. finetianum*, and *A. sinomontanum*.** The genes lying inside and outside the circles are transcribed in the clockwise and counterclockwise directions, respectively. Different colors denote the genes belonging to different functional groups. The thicknesses indicate the extent of the inverted repeats (IRa and IRb) that separate the small single-copy (SSC) region from the large single-copy (LSC) region. The dark gray in the inner circle corresponds to GC content, and the light gray to AT content.

When duplicated genes in the IR regions were counted only once, each of the three cp genomes encode 126 predicted functional genes, including 84 PCGs, 38 tRNA genes, and four rRNA genes. The remaining non-coding regions include introns, intergenic spacers, and pseudogenes. Altogether 18 genes were duplicated in the IR regions, including seven PCGs, seven tRNA genes, and four rRNA genes (Fig. 1; Table S1). Each of the thirteen genes (eight PCGs and five tRNA genes) contained one interval, and three PCGs (*clp*P, *ycf*3 and *rps*12) had two intervals each (Table S1). The maturase K (*mat*K) gene in the cp genomes of the three species is located within *trn*K intron, which is similar in most of the other plants species (*Kong & Yang, 2017*). In the IR regions, the four rRNA genes and two tRNA genes (*trn*I and *trn*A) are clustered as 16S-*trn*I-*trn*A-23S-4.5S-5S. This has also been

reported in the cp genomes of *A. barbatum* var. *hispidum*, *A. barbatum* var. *puberulum*, and many other plant species (*Mardanov et al., 2008*; *Wu et al., 2014*; *Chen et al., 2015*).

## Comparative analysis of genomic structure

Synteny analysis identified a lack of genome rearrangement and inversions in the cp genome sequences of the *Aconitum* species. No gene rearrangement and inversion events were detected (Fig. S1). Genomic structure, including gene number and gene order, is highly conserved among the *Aconitum* species; however, structural variation was still present in the LSC/IR/SSC boundaries (Fig. 2). The genes *rps*19-*rp*12-*trn*H and *ycf*1-*ndh*F were located between the junction of the LSC/IR and SSC/IR regions. The *rps*19 gene crosses the LSC/IRa junction region in *A. sinomontanum*, *A. barbatum* var. *puberulum* and *A. barbatum* var. *hispidum* of subg. *Lycoctonum*, as well as in *A. jaluense*, *A. volubile*, *A. carmichaelii*, *A. kusnezoffii* and *A. monanthum* of subg. *Aconitum*. As a result, the *rps*19 gene has apparently lost its protein-coding ability due to being partially duplicated in the IRb region, thus a producing pseudogenized Ψ*rps*19 gene. The same was found with the *ycf*1 gene, as the IRb/SSC junction region is located within the *ycf*1 CDS region and only a partial gene is duplicated in the IRa region, resulting in a pseudogene. This is a general structure among the dicots. The Ψ*ycf*1 pseudogene in the IR region was 1,279 bp for two varieties in subg. *Lycoctonum* and seven species in subg. *Aconitum*. However, length variation was present in the IR of the remaining six species: 1,292 bp in *A. angustius*, *A sinomontanum*, and *A. reclinatum*; 1,165 bp in *A. finetianum*; 1,274 bp in *A. chiisanense*; 1,356 bp in *A. volubile*; and 1,263 bp in *A. carmichaelii* (Fig. 2; Table 2).

Three pseudogenes, Ψ*ycf*15, Ψ*rps*16, and Ψ*inf*A, were identified in the gene annotations (Table 2). The Ψ*ycf*15 gene is pseudolized in *A. austrokoreense* and *A. chiisanense* with four base insertions and pseudolized in *A. monanthum* with a one base insertion, contributing to several internal stop codons. The Ψ*inf*A region is pseudogenized with two nonsynonymous substitutions producing internal stop codons in all of the members of subg. *Lycoctonum*. This pseudogenized Ψ*inf*A gene has also been found in other angiosperm chloroplast genomes (*Roman & Park, 2015*; *Lu, Li & Qiu, 2017*). The gene *rps*16 encodes the ribosomal protein S16 and is present in the cp genome of most if the higher plants. However, *rps*16 has been functionally lost in various plant species (*Roy et al., 2010*). A pseudogene Ψ*rps*16 was also present in the cp genomes of *A. angustius*, *A. finetianum* and *A. reclinatum* in subg. *Lycoctonum* as well as in the nine species in subg. *Aconitum* due to the loss of one CDS region (Table 2). As has been revealed in other studies, the functional loss of the *rps*16 gene might be compensated by the dual targeting of the nuclear *rps*16 gene product (*Keller et al., 2017*).

## Sequence divergence among the species in *Aconitum*

The average nucleotide variability (*Pi*) values were estimated to be 0.00549, ranging from 0 to 0.03856, based on the comparative analysis of cp genome sequences in *Aconitum* species. The highest variation was found in the LSC and SSC regions, with an average $Pi = 0.007140$ and 0.008368, respectively. The IR regions had a much lower nucleotide diversity with

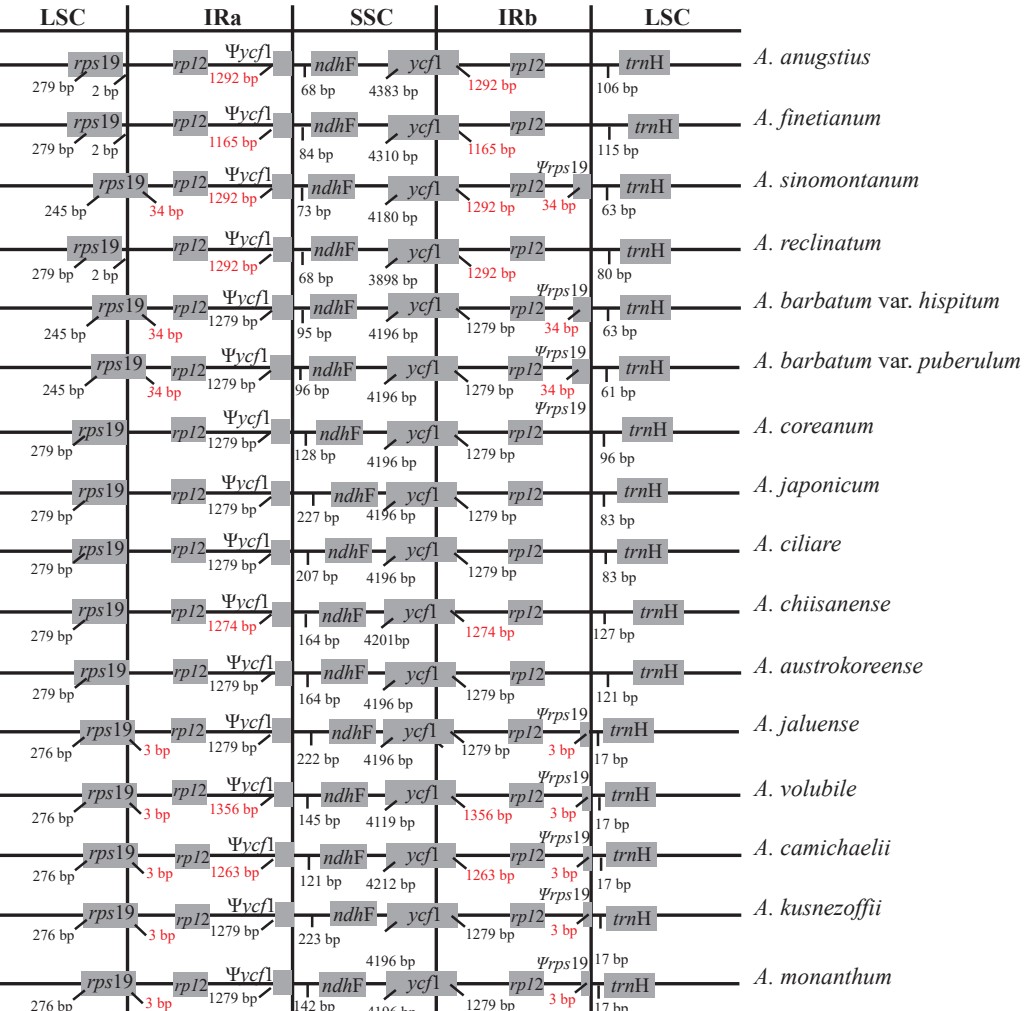

**Figure 2 Comparison of the border positions of LSC, SSC and IR repeat regions among fourteen species and two varieties in *Aconitum*.** Genes are denoted by grey boxes and the gaps between the genes and the boundaries are indicated by the base lengths (bp). Extensions of the genes are also indicated above the boxes.

$Pi = 0.001079$ and $0.001459$. Eight intergenic regions (*trn*H-*psb*A, *trn*K-*rps*16, *trn*D-*trn*Y, *trn*Y-*trn*E, *trn*E-*trn*T, *trn*T-*trn*L, *rpl*12-*clp*P and *trn*H-*trn*R) were highly variable, with *Pi* value ~0.023 (Fig. 3). The former eight loci are present in the LSC, while the pseudogene Ψ*ycf* 1 is in the SSC region. The single-copy regions have been demonstrated to be highly variable with loci clustered in 'hot spots' (*Kong & Yang, 2017*). Among the eight intergenic regions, *trn*H-*psb*A and *trn*T-*trn*L are variable and useful for phylogenetic reconstruction in the subg. *Lycoctonum* (*Utelli, Roy & Baltisberger, 2000*; *Kong et al., 2017b*). However, the other intergenic regions, even with higher nucleotide variability, have never been involved in the phylogenetic analysis for the genus *Aconitum*. The highly variable loci detected in the current study may provide a basis for further phylogenetic characterization of this

**Table 2  The distribution of the five pseudogenes in *Aconitum*.**

| Locations | LSC | | LSC/IRa | IRa | IRa/SSC |
|---|---|---|---|---|---|
| Genes | Ψ*rps*16 | Ψ*inf*A | Ψ*rps*19 | Ψ*ycf*15 | Ψ*ycf*1 |
| ***Aconitum* subg. *Lycoctonum*** | | | | | |
| *A. angustius* | + | | | | +/1,292 bp |
| *A. finetianum* | + | | | | +/1,165 bp |
| *A. sinomontanum* | | + | +/34 bp | | +/1,292 bp |
| *A. reclinatum* | + | + | | | +/1,292 bp |
| *A. barbatum* var. *puberulum* | | + | +/34 bp | | +/1,279 bp |
| *A. barbatum* var. *hispidum* | | + | +/34 bp | | +/1,279 bp |
| ***Aconitum* subg. *Aconitum*** | | | | | |
| *A. austrokoreense* | + | | | +/4 bp indel | +/1,279 bp |
| *A. carmichaelii* | + | | +/3 bp | | +/1,263 bp |
| *A. chiisanense* | + | | | +/4 bp indel | +/1,274 bp |
| *A. ciliare* | + | | | | +/1,279 bp |
| *A. coreanum* | + | | | | +/1,279 bp |
| *A. jaluense* | + | | +/3 bp | | +/1,279 bp |
| textit*A. japonicum* | + | | | | +/1,279 bp |
| *A. kusnezoffii* | + | | +/3 bp | | +/1,279 bp |
| *A. monanthum* | + | | +/3 bp | +/1 bp indel | +/1,279 bp |
| *A. volubile* | + | | +/3 bp | | +/1,356 bp |

**Notes.**

+, indicating the presence of pseudogenes.

genus. The observed divergence hotspot regions provide abundant information for marker development in phylogenetic analysis or conservation genetics of *Aconitum*.

## Characterization of simple sequence repeats

MISA was used to identify SSRs with minimum a of 10 bp repeats among the three species. In *A. angustius*, 60 SSRs were found, while 62 SSRs were found in *A. finetianum*, and 58 in *A. sinomontanum*. This result is comparable with *A. reclinatum* (61 SSRs), *A. barbatum* var. *hispidum* (53 SSRs), and *A. barbatum* var. *puberulum* (57 SSRs). An average of 59 SSRs were identified in subg. *Lycoctonum*, which is relatively higher than that of subg. *Aconitum* (47). In both subgenera, most SSRs are in the LSC regions, accounting for an average of 85.31% and 80.85% in subg. *Lycoctonum* and subg. *Aconitum*, respectively. Among all of the SSRs, the mononucleotide A/T repeat units occupied the highest proportion, with 64.41% and 46.82% of the total SSRs in subg. *Lycoctonum* and subg. *Aconitum*, respectively. Although few SSRs were detected in subg. *Aconitum*, a higher proportion of di-, tri-, tetra- and penta-nucleotide repeats were detected (Table 3). The SSRs have a remarkably high A/T content with only seven SSRs, namely $(ATCT)_3$, $(TTCT)_3$, $(CTTT)_3$, $(TAAAG)_3$, $(TTTC)_3$, $(ATAC)_3$ and $(CATT)_3$, that contain one C or G nucleotide.

A total of 11 cp SSR loci were shared among the cp genomes of tetraploid *A. angustius* and diploid *A. finetianum*. No common cp SSRs were specifically found between *A. angustius* and *A. sinomontanum*. This result provides evidence of the maternal origin of the tetraploid *A. angustius* from diploid *A. finetianum*, which is consistent with previous research

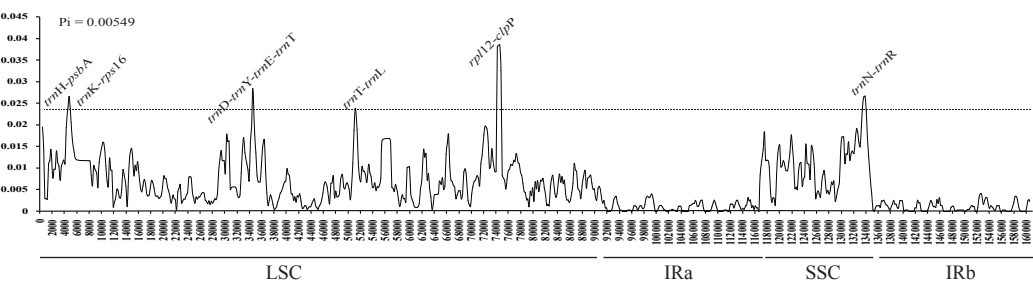

**Figure 3** **Sliding window analysis of the whole cp genome for fourteen species and two varieties in**
***Aconitum.*** *X*-axis, position of the midpoint of a window; *Y*-axis, nucleotide diversity (*Pi*) of each window.

(*Gao, 2009*; *Kong et al., 2017b*). Among the three species, the highest number of unique SSRs loci were present in *A. sinomontanum* (11) followed by *A. angustius* (7), *A. finetianum* (6), and *A. reclinatum* (5).

## Phylogenetic analyses

In the present study, three phylogenetic methods (BI, MP and ML) resulted in identical phylogenetic trees within each data set. Different analyses based on the two datasets generated largely congruent topologies (Fig. 4). The total aligned length with parsimony informative loci was 178,392 bp with 4,342 for the complete cp genome sequences, and 106,535 bp with 3,164 for PCGs, respectively. All of the phylogenetic trees support that *Aconitum* comprises two monophyletic subgenera. High Bayesian posterior probabilities and bootstrap values were detected at most nodes, particularly based on the complete cp genomes (Fig. 4A).

The phylogenetic relationship of Korean species in subg. *Aconitum* was investigated for the first time. The monophyletic clade was formed by *A. ciliare*, *A. carmichaelii*, *A. japonicum* subsp. *napiforme* and *A. kusnezoffii*, with strong support values (Fig. 4). The clade comprised of *A. jaluense* subsp. *jaluense* and *A. volubile* exhibited moderate-to-high support, forming a monophyletic sister group. The positions of the four species *A. ciliare*, *A. carmichaelii*, *A. austrokoreense*, and *A. chiisanense*, demonstrated inconsistencies based on the two data sets. Obviously, these species received stronger support based on the sequences of the complete cp genome rather than PCGs, indicating that whole genomes are more efficient in determining phylogenetic relatedness in *Aconitum* than PCGs alone.

Based on the phylogenetic tree, the tetraploid *A. angustius* was always closely related with diploid *A. finetianum*, which further supports previous research (*Kong et al., 2017b*). The two species co-occur on several mountains in southeast China and even grow very closely within a community (*Yuan & Yang, 2006*). They show similar morphological characteristics in having three-part leaves, the cylindric upper sepals and retrosely pubescent pedicels, resulting in common misidentification (*Gao, Ren & Yang, 2012*). *Aconitum finetianum* is the most likely maternal progenitor of *A. angustius* based on both molecular phylogenetic and morphological evidence (*Kong et al., 2017b*); therefore, it is reasonable to see that the two species have a close phylogenetic relationship.

**Table 3** Number of chloroplast SSRs in different regions or different types present in *Aconitum* species.

| Species | Homo (>10) | Di (>5) | Tri (>5) | Te(>3) | Pen (>3) | Number of SSRs in different regions | | | |
| --- | --- | --- | --- | --- | --- | --- | --- | --- | --- |
| | | | | | | LSC | SSC | IR | Total |
| subg. *Lycoctonum* | 38 (64.41%) | 10 (16.95%) | 3 (4.80%) | 8 (12.99%) | 0 (0.00%) | 50 (85.31%) | 7 (11.02%) | 2 (3.39%) | 59 |
| *A. angustius* | 40 | 9 | 2 | 8 | 1 | 50 | 8 | 2 | 60 |
| *A. finetianum* | 42 | 9 | 2 | 8 | 1 | 51 | 9 | 2 | 62 |
| *A. sinomontanum* | 36 | 12 | 2 | 8 | 0 | 50 | 6 | 2 | 58 |
| *A. reclinatum* | 42 | 10 | 2 | 7 | 0 | 53 | 6 | 2 | 61 |
| *A. barbatum* var. *puberulum* | 36 | 10 | 2 | 8 | 0 | 49 | 5 | 2 | 56 |
| *A. barbatum* var. *hispidum* | 32 | 10 | 7 | 7 | 0 | 49 | 5 | 2 | 56 |
| subg. *Aconitum* | 22 (46.81%) | 15 (31.91%) | 1 (21.28%) | 7 (14.89%) | 1 (21.28%) | 38 (80.85%) | 7 (14.89%) | 2 (4.36%) | 47 |
| *A. austrokoreense* | 22 | 15 | 0 | 7 | 0 | 32 | 10 | 2 | 44 |
| *A. carmichaelii* | 21 | 16 | 1 | 7 | 0 | 37 | 6 | 2 | 45 |
| *A. chiisanense* | 21 | 16 | 1 | 7 | 2 | 39 | 6 | 2 | 47 |
| *A. ciliare* | 23 | 16 | 1 | 7 | 1 | 41 | 5 | 2 | 48 |
| *A. coreanum* | 39 | 14 | 1 | 7 | 1 | 50 | 10 | 2 | 62 |
| *A. jaluense* | 17 | 14 | 1 | 6 | 2 | 33 | 6 | 2 | 41 |
| *A. japonicum* | 20 | 16 | 1 | 7 | 1 | 37 | 6 | 2 | 46 |
| *A. volubile* | 17 | 15 | 1 | 6 | 1 | 35 | 3 | 2 | 40 |
| *A. kusnezoffii* | 19 | 16 | 1 | 7 | 1 | 37 | 5 | 2 | 44 |
| *A. monanthum* | 18 | 13 | 0 | 7 | 2 | 36 | 9 | 2 | 47 |

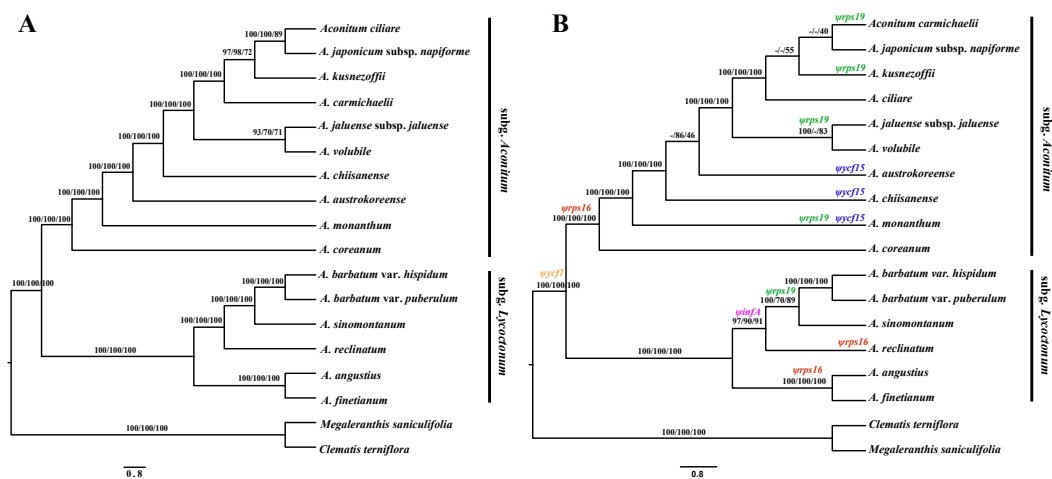

**Figure 4** **Phylogenomic relationship among *Aconitum* species.** Based on the two data sets of complete cp genome sequences (A) and PCGs (B), respectively, phylogenetic reconstruction was conducted using three methods: Bayesian Inference (BI), Maximum Parsimony (MP) and Maximum Likelihood (ML). Numbers above the branches represent BI posterior probabilities, MP and ML bootstrap values. The pseudogenes are indicated above the branches in different colors on the phylogenetic tree based on PCGs (B).

The five pseudogenes exhibit different evolutionary histories from each other. Concerning the evolution of Ψ*ycf15*, it occurs in only three species *A. monanthum, A. austrokoreense*, and *A. chiisanense* of subgen. *Aconitum*, which was probably pseudogenized once in each species independently and subsequently restored to a functional copy. We propose that Ψ*rps16* was pseudogenized during the divergence between the two subgenera and restored to a functional copy within the *A. sinomontanum-A. barbatum* clade of subgen. *Lycoctonum*. With respect to Ψ*rps19*, it appears to have been pseudogenized multiple times independently in phylogenetically distant species of the two subgenera. Ψ*ycf1* is commonly found among cp genomes of plant species. Within *Aconitum*, Ψ*ycf1* exhibits length variation and multiple convergent mutation events, which are not consistent with the phylogenetic relationships of the genus. Only Ψ*infA* shows an evolutionary history congruent with the phylogeny of *Aconitum* (Fig. 4B; Table 2). Overall, our results show that similarities among pseudo-gene sequences do not necessarily predict phylogenetic relationships among species.

## ACKNOWLEDGEMENTS

We would like to provide a special thank to Dr. Tongjian Liu for his assistance in lab work and data analyses. We thank Dr. AJ Harris and LetPub (http://www.letpub.com) for their linguistic assistance during the preparation of this manuscript.

### Funding

This work was supported by the National Natural Science Foundation of China (31470319; 31470312) and Science and Technology Planning Project of Guangdong Province, China (2016A030303048; 2017A030303067). The funders had no role in study design, data collection and analysis, decision to publish, or preparation of the manuscript.

### Grant Disclosures

The following grant information was disclosed by the authors:
National Natural Science Foundation of China: 31470319, 31470312.
Science and Technology Planning Project: 2016A030303048, 2017A030303067.

### Competing Interests

The authors declare there are no competing interests.

### Author Contributions

- Hanghui Kong and Wei Gong conceived and designed the experiments, analyzed the data, wrote the paper, prepared figures and/or tables, reviewed drafts of the paper.
- Wanzhen Liu performed the experiments, analyzed the data, contributed reagents/materials/analysis tools.
- Gang Yao wrote the paper, prepared figures and/or tables, reviewed drafts of the paper.

### DNA Deposition

The following information was supplied regarding the deposition of DNA sequences:
Accession data available from GenBank (MF155664, MF155665 and MF155666) and from figshare (https://dx.doi.org/10.6084/m9.figshare.5092414.v1).

### Data Availability

Raw data for whole cp genome matrix: Kong, Hanghui (2017): cp_whole.nex. figshare. https://dx.doi.org/10.6084/m9.figshare.5092414.v1.
Raw data for CDs matrix: Kong, Hanghui (2017): CDS.fas.nex. figshare. https://dx.doi.org/10.6084/m9.figshare.5092420.v1.
The of cp genome sequences are contained at the GenBank accession numbers in Table 1.

### Supplemental Information

Supplemental information for this article can be found online at http://dx.doi.org/10.7717/peerj.4018#supplemental-information.

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
