# Peer review of "A comparison of chloroplast genome sequences in Aconitum (Ranunculaceae): a traditional herbal medicinal genus"

_PeerJ, doi:10.7717/peerj.4018_

## Round 0.1 · original submission · Major Revisions

Dear authors

As you see , our reviewers ask for a thorough revision. From my side, I would ask you to include more Aconitum species in your phylogenetic analysis; this genus is also widely distributed in Europe and these sequences are in Genbank but not included in your analysis.

A good phylogeny should be based of a selection of taxa which should be as complete and representative as possible.

Your revision will be reviewed again.

Greetings

Michael Wink
Academic editor

·

Basic reporting

The manuscript is generally well written and, apart from some minor grammatical errors and ambiguous sentences as indicated below, the language used is clear to understand.
The introduction section has effectively been written, introducing all the aspects of the study and referencing previous studies on genus Aconitum. However, there is need for more details in order to support the need for further molecular systematics research.
The two most recent studies by Hong et al. (2017) and Kong et al. (2017) on phylogenetic reclassification (reconstruction) appear to have clearly outlined phylogenetic relationships within the genus Aconitum by using multiple nuclear and chloroplast markers.
The authors in the current study acknowledge the effectiveness of universal markers of cp genome to resolve phylogenetic relationships among species (line 64-68). However, they claim that there is need for further research using data obtained from whole genome (Phylogenomics). To support their claim the authors have stated in line 94-95 and 99-100, that there are some species whose positions are yet to be resolved. The authors should list examples of a few taxa whose phylogenetic positions are unclear and outline how the current data will resolve.
The cited literature is relevant and recent.
The reporting structure is in line with the standards of the journal.
Figures are well presented and of good quality.
Information on raw data deposition and availability has been given by the authors.

Experimental design

The design used here seems sound, replicable and in line with the research concerns highlighted in the introduction section. The research is within the scope of the journal.

Validity of the findings

The results presented here are reliable based on the experimental design. The results and discussion section is sound and the readers can be able to draw conclusions based on the results discussed here.

Additional comments

The manuscript is well presented. However, to ensure that international readers understand the intended meaning in some statements, I suggest you further review the manuscript. Below, and as highlighted in the attached pdf, I have listed some of the points that may need your attention. Please compare the suggested statements to the original ones.
Line 32
The length of the cp genome sequences was 156,109 bp for A. angustius, 155,625 bp for A. finetianum and 157,215 bp for A. sinomontanum.

Line 33
Each of the three species possesses 126 genes including 84 protein coding genes (PCGs).

Please use ‘’structural variations’’ instead of ‘’structure variations’’ to refer to changes in genome organization.
Line 40
In total, 50 simple sequence repeats (SSRs) were detected in A. finetianum and A. angustius, while 57 SSRs were discovered in A. sinomontanum.

43
A higher percentage of…

45
..will be of benefit for further/ will benefit further….

54
The cp genome in angiosperms is a circular DNA molecule with typically quadripartite structure

74
…possesses heterologous chromosomes and is suggested to be a hybrid of A. finetianum and A. sinomontanum.

80
In the past decades, the genus Aconitum was known as a taxonomically and phylogenetically challenging taxon.

86
…ITS, which suggested Aconitum to be a monophyletic clade and a sister group of Delphinium

103
We further compared

114
Fresh leaves were collected from A. angustius, A. finetianum and A. sinomontanum growing in a greenhouse at South China Botanical Garden, Chinese Academy of Sciences.

124
Genomic DNA was fragmented randomly and then the required length DNA fragments were retained by electrophoresis


The minimum numbers (thresholds) of the SSRs were set to be 10, 5, 4, 3, and 3 for mono-, di-, tri-, tetra-, and penta-nucleotides SSRs respectively.

134 comments
You sequenced and annotated three species, here you report four species. Please confirm if this is correct.

OGDRAW should be in parenthesis

194
…. by manually identifying the overlapping regions.

195
In order to further determine the draft genome, quality and coverage of each base position by reads remapping were double checked and corrected accordingly.

200
The chloroplast genomes displayed a typical quadripartite structure, including a pair of IRs (25,927-26,225 bp) separated by LSC (86,664-88,074 bp) and SSC (16,914-17,107 bp) regions (Fig. 1 and Table 1).

202-205
I suggest you use 1 decimal place instead of 2.
The GC content of the three species was 38.0%, demonstrating congruence to that reported in A. barbatum var. hispidum and A. barbatum var. puberulum (38.0%) of subg. Lycoctonum as well as in the species of subg. Aconitum (38. 0% or 38.1%) (Table 1).

206
Please avoid listing the names of the species again, when it is clear that you’re referring to them.
When duplicated genes in IRs regions were counted only once, each of the three cp genomes encoded 126 predicted functional genes, including 84 PCGs, 38 tRNA genes and four rRNA genes.

212
In the three studied species, and similar to most other plants species, the maturase K (matK) gene is located within trnK intron.

213
In the IR regions, the four rRNA genes and two tRNA genes (trnI and trnA) are clustered as 16S-trnI-trnA-23S-4.5S-5S. This is also reported in the cp genomes of A. barbatum var. hispidumand and A. barbatum var. puberulumas as well as in many other plant species

223-225
However, structural variations were still present in the LSC/IR/SSC boundaries (Fig. 2). The genes rps19-rp12-trnH and ycf1-ndhF were located in the junction regions of LSC/IR and SSC/IR respectively.

225-230
The rps19 gene crosses the LSC/IRa junction in A. sinomontanum, A. barbatum var. puberulum and A.barbatum var. hispidum of subg. Lycoctonum, as well as in A. jaluense, A. volubile, A.carmichaelii, A. kusnezoffii and A. monanthum of subg. Aconitum. As a result, the gene has apparently lost its protein-coding ability due being partially duplicated in the IRb region, thus producing pseudolized Ψrps19 gene.

253
The average nucleotide variability (Pi) value was estimated to be 0.00549, ranging from 0 to 0.03856, based on comparative analysis of sequence divergence of complete cp genome sequences of Aconitum species.

271
MISA identified 50 SSRs, with minimum 10 bp repeats in length, in both A.
finetianum and A. angustius, while 57 SSRs were detected in A. sinomontanum.

280
Remarkably the SSRs had a high A/T content with only seven SSRs, including (ATCT)3, (TTCT)3, (CTTT)3, (TAAAG)3, (TTTC)3, (ATAC)3 and (CATT)3, containing one C or G nucleotide.

294
The total aligned length with parsimony informative loci was 178,392 bp with 4,342 for the complete cp genome sequences, and 106,535 bp with 3,164 for PCGs, respectively.

300
Based on the phylogenetic tree, the tetraploid A. angustius was always closely related to A. finetianum, which is also supported by previous research


Figure 1.
One of the ycf15 duplicates, on the IR, is colour coded as ‘other genes’ while in table two the gene is listed under ‘’genes for photosynthesis’’. Please check alongside the partially duplicated gene of ycf1.Probably this is because the duplicates have been annotated on the same strand of the DNA molecule.

Figure 2.
Please check the use of ‘inverted’ in the title. On the same, kindly consider using ‘’IR regions’’ or just ‘IRs’ while referring to the two IRs rather than using ‘IRs regions’.

·

Basic reporting

Please have a native English speaker revise the writing. Examples of mistakes include lines 26-27, 70, 80-81, 86, 107-108, 305. There are many others, these are some of the worst.

Experimental design

No herbarium specimen vouchers are cited. Many journals now require this for publication. Vouchers are essential for reproducibility, especially in taxa where identification is challenging.
Considering that chromosome number is the best way to distinguish two of the species, ploidy confirmation is also desirable.
Line 189 describes results from de novo assembly, while line 129 describes a reference mapping method. Which is correct?
The phylogenetic analysis describes 3 methods used, but only 2 trees are shown (Fig. 4A & B). The nodal support values are not identified.
Lines 295-297: The 2 trees shown are not “mostly concordant” since 4 nodes differ, and 3 nodes in 4B do not have high support values.
Figure 1 is not necessary, since the difference between the plastomes is shown in Figure 2. All other aspects of gene organization do not differ from published Ranunculaceae.
Table 2 is not necessary, since this is the typical angiosperm gene content and structure.

Validity of the findings

Most of the plastomes used in the comparative analyses are unpublished sequences downloaded from GenBank, with no attribution other than citing accession numbers. What evidence do the authors have that these unpublished plastomes are properly identified, assembled and annotated?
The comparative analyses fail to incorporate the Aconitum reclinatum plastome, recently published by the same authors. This is unjustified.
My recommendation is to remove the phylogenetic analysis, since it does not provide much new information beyond the results already published in the A. reclinatum paper, and limit the gene content/structure analysis to species with reliable published plastomes.

---

## Round 0.2 · Major Revisions

Dear authors

Please take the recommendations of the reviewer seriously. Otherwise, we cannot accept your ms

Kind Regards
Michael Wink

·

Basic reporting

no comment

Experimental design

no comment

Validity of the findings

no comment

·

Basic reporting

I had recommended removing the phylogenetic analysis – but the authors kept it in the manuscript. If it is retained, I insist that the authors fully acknowledge the source of all chloroplast genomes, including those that are unpublished. I am in favor of open data, but this also requires full attribution.

Two citations were added for published plastomes (Lim et al., 2015; Choi et al., 2016) but these are not included in the literature cited.
The unpublished Aconitum plastomes from Korea need to include a citation of the authors listed in the Genbank records: “Kim, G.-B., Lim, C.-E. & Mun, J.-H. unpublished. Complete chloroplast genomes of Aconitum species from Korea. Obtained from Genbank on Date”

Voucher numbers have been added (from Genbank records) for the source specimens of the plastome sequences, but these are incomplete – we also need to know the herbarium where the voucher is stored. This is missing from Genbank and will need to be obtained from the submitters of those sequences.

The phylogenetic distribution of the pseudogenes should be described (now that the phylogenetic analysis is retained). Interestingly, only pseudo-infA corresponds to a clade (i.e. single pseudogenization). In the case of pseudo-ycf15 the 3 species A. monanthum, austrokoreense & A. chiisanense are not monophyletic. This suggests the tree is incorrectly resolved – or less likely, the gene was lost once and then restored to a functional copy. These possibilities need to be discussed. Likewise, rpl16 was either pseudogenized twice, or restored to function once in subg. Lycoctonum.

“pseudolized” is not a word. Replace with pseudogenized.

Experimental design

no comment

Validity of the findings

no comment

---

## Round 0.3 · Minor Revisions

Please make the corrections to the English which are required by the reviewer.

·

Basic reporting

The added text on the phylogeny of pseudogenes (lines 360-370) requires substantial English improvement.

Experimental design

no comment

Validity of the findings

no comment

Additional comments

The authors have adequately addressed my concerns.

---

## Round 0.4 · accepted · Accept

Congratulations- now, your ms can be accepted

Greetings
Michael Wink
Academic editor